# Mechanistic Paradigms of Natural Plant Metabolites as Remedial Candidates for Systemic Lupus Erythromatosus

**DOI:** 10.3390/cells9041049

**Published:** 2020-04-22

**Authors:** Acharya Balkrishna, Pallavi Thakur, Shivam Singh, Swami Narsingh Chandra Dev, Anurag Varshney

**Affiliations:** 1Drug Discovery and Development Division, Patanjali Research Institute, NH-58, Haridwar, Uttarakhand 249 405, India; 2Department of Allied and Applied Sciences, University of Patanjali, Patanjali Yog Peeth, Roorkee-Haridwar Road, Haridwar, Uttarakhand 249 405, India

**Keywords:** autoimmunity, immune regulation, natural plant products, systemic lupus erythematosus, herbal medicines

## Abstract

Systemic lupus erythematosus (SLE) is a complex autoimmune disorder involving a dysregulated immune response which ultimately leads to multiple organ failure. Several immunological and cellular checkpoints are available as drug targets. However, the available chemosynthetic drugs such as non-steroidal anti-inflammatory drugs and corticosteroids provide limited therapy with extreme toxicities. Moreover, the disease heterogeneity in SLE is very difficult to manage by a single drug component. Hence, it is imperative to utilize the holistic capabilities of natural plant products as immunomodulators and intracellular signaling regulators, thereby providing an auxiliary option of treatment. Additionally, the herbal drugs also serve as symptomatic relief providers, thereby serving as a prophylactic remedy in case of cerebrovascular, hepatic, nephropathological, hematological, cardiopulmonary, mucocutaneous and musculoskeletal manifestations of SLE. The present review attempts to showcase the current state of knowledge regarding the utility of plant-derived phyto-metabolites with their probable mechanistic roles in treating SLE, by means of targeting the signaling cascade, proinflammatory cytokine production and B–T cell co-stimulation. It is hoped that further preclinical and clinical studies will be embarked upon in order to understand the underlying therapeutic and mechanistic aspects of these medicinal herbs.

## 1. Introduction

Systemic lupus erythromatosus (SLE) is a chronic, incapacitating autoimmune disorder with an unusual trend of latency and relapses, mainly afflicting women of Asian, African or American ethnicities [1]. The unpredictable course of relapsing and remissions makes it difficult to diagnose [2]. Moreover, there is no absolute treatment regime for this ailment. However, certain immunosuppressive steroids offer a limited therapeutic way of providing symptomatic relief. As a consequence, these steroids often render the SLE patients susceptible to opportunistic infections [3,4].

The main etiological factors responsible for causing SLE primarily include genetic factors (presence of inflammation triggering genes, toll-like receptor expressing genes and STAT signaling pathway genes) [1,5,6]. The genetic structures of SLE-responsive individuals have also exhibited the absence/masking of DNA repair genes such as *TREX1*, *KLK1* and *KLK3* [7,8,9].

Certain epigenetic changes such as post-translation modification of histones and DNA methylation often induce susceptibility to SLE. External factors such as ultraviolet rays, demethylating drugs (Azacitidine, Decitabine) and viral infection (Epstein–Barr virus) often trigger the possibility of occurrence of SLE (Figure 1). Moreover, the higher concentration of hormones like estrogen and prolactin also leads to the upregulation of autoimmune phenotypes and hence, SLE is associated with the female gender more often. Both hormones promote the chances of survival of plasma cells responsible for producing high affinity autoreactive B cells [10,11].

All these etiological factors lead to certain immune dysfunctionalities (defective neutrophils, macrophages and natural killer cells), thereby causing the immune system to attack its own cells and destroy multiple organ systems including the cardiovascular, mucocutaneous, cerebrovascular and pulmonary systems, etc., ultimately leading to SLE [8]. Characteristic symptoms of the disease include maculopapular (e.g., malar or discoid, butterfly shaped facial and cutaneous rashes), mucocutaneous (oral ulcers), musculoskeletal (arthritis and osteomyelitis), cerebrovascular (cranial neuropathies and cognitive impairments), hepatic (hepatomegaly, cholecystitis), renal (nephritis, proteinuria) and cardiopulmonary manifestations (pericarditis, pneumonitis). Autoantibodies against the nuclear body, ribonucleoproteins and DNA are formed in SLE. Moreover, the immunological and hematopoietic cells of the body are also attacked by autoantibodies, thereby causing severe damage to the blood cells as reflected by hemolytic anemia, leukopenia and thrombocytopenia [17]. Further secondary manifestations are also observed wherein the vital organs such as the liver, heart, lungs and kidneys undergo acute failure, thereby increasing morbidity and mortality (Figure 2).

Most of the clinical manifestations of the disease are associated with certain pathophysiological cascades, particularly involving the hyperactivation of the immune response and abnormality in the immune regulation system. Proinflammatory cytokines, namely IL6, IL10, IL12, IL17, IL21 and IL23, are produced in excess as mediated by rho associated protein kinase (ROCK); transcription factors, namely STAT3 and CREMα; nuclear factor NFκB and NFAT, thereby leading to the co-stimulation of B and T cells, ultimately causing the excessive production of autoantibodies. Furthermore, transcription and nuclear factors, such as CREB, NFκB and NFAT also lead to the production of BCL6 which downregulates the production of immunoregulatory cytokines, i.e., IL2 [12,13,14,15,16,17,18]. Excessively produced autoantibodies lead to DNA and ribonucleoprotein damage in healthy cells, leading to mass apoptosis. Hence, both autoantibodies and inflammatory cytokines are responsible for aggravating the pathology of SLE [13,14].

This disease follows an erratic course of latency and recurrence, thereby making its prognosis and diagnosis difficult [2]. Current therapeutic strategies of this disease are also limited to the use of steroids and cytotoxic drugs. These remedial regimes also possess the limitations of exorbitant costs and the induction of multiple organ toxicity. Hence, SLE can be designated as a life-threatening disease wherein diagnosis and therapeutic challenges still exist. There is an urgent need for the investigation on better therapeutic approaches for targeting the etio-pathogenic mechanisms of SLE [3]. The toxicological profiles of currently available chemotherapeutic drugs advocate the exploratory search for natural remedial sources for the management of SLE. As a matter of fact, combinatorial therapies of chemosynthetic drugs and holistic nutraceuticals can also be examined for assessing the possibilities of immuno-augmentation [22]. The following sections present the mechanistic roles of phytoconstituents with their natural sources as promising remedial targets for various pathophysiological checkpoints of SLE (Table 1).

## 2. Chemotherapeutic Modalities for Treating SLE

The primary remedial regimes for SLE include corticosteroidal drugs, non-steroidal anti-inflammatory drugs (NSAIDs) and antimalarial drugs. These remedies are not absolute but are the only available option for curbing this disease and its related symptoms. Corticosteroids are frequently used to suppress the acute, inflammatory condition of SLE. They are prescribed in either high or low doses for treating severe to long-term manifestations. Unfortunately, a significant proportion of patients still depend on corticosteroids for months to a lifetime. Such dependency on steroids has become a grave clinical problem, wherein any attempt to reduce or discontinue the dose of corticosteroids further aggravates the complications [43]. Moreover, there are several adverse effects of corticosteroids, such as acne, edema, hirsutism, myopathy, hypertension, cataracts and diabetes [44]. In certain cases, prolonged dependency on corticosteroids causes avascular necrosis, osteonecrosis [44,45] and microvascular tamponade from hypertrophy of intraosseous lipocytes [44,46,47].

Besides, nonsteroidal anti-inflammatory drugs (NSAIDs) are universally prescribed to relieve arthralgia, inflammation, serositis and fever in SLE patients [48]. NSAIDs are blockers of cyclooxygenases, the COX-1 and COX-2 enzymes. Such selective inhibition of COX-1 and COX-2 facilitates the production of prostaglandins that reduce inflammation and pain [49,50,51]. However, these NSAIDs often impart gastrointestinal side effects such as adverse bleeding and antiplatelet effects. Hence, the use of NSAIDs for SLE requires strict caution when prescribed to patients with renal disease or photosensitivity [51].

Furthermore, most of the patients with SLE are advised to administer antimalarial drugs (hydroxychloroquine and chloroquine) as auxiliary treatment regimens [8,52]. The mode of action of antimalarial drugs is to inhibit the transcription of intracellular toll-like receptors, thereby ameliorating the autoimmune response [52,53,54]. Additionally, the antimalarials aid in managing the disease conditions of SLE, such as skin lesions, neoplasm and thromboembolism. However, rampant use of antimalarials often leads to renal and hepatic complications [54].

It is noteworthy that the monoclonal antibody biologic, rituximab, has been investigated for its efficacy against SLE as it induces B cell depletion, thereby controlling the overproduction of autoantibodies [55]. Moreover, the B cell hyperactivity in SLE is associated with the over-expression of a B cell regulator, namely B-cell activity factor (*BAFF*), which can be downregulated by employing anti-BAFF antibodies [56]. 

## 3. Plant-Derived Natural Compounds for Treating SLE

A number of plant-derived phytoconstituents are known to exert immunosuppressive effects and hence can be used for the management of autoimmune disorders such as SLE. Phytocompounds belonging to diverse categories as alkaloids (berbamine), lectin (tomato lectin), flavonoids (quercetin), phenolic glycosides (curcumin), terpenoids (azadirachtin) and saponins (ginsenoside) have been found to reduce inflammatory cytokines and intracellular signaling, thereby exerting anti-inflammatory and immunoregulatory effects [15,35]. Moreover, certain natural plant products (e.g., pyrogallol, cynocobalamin, folacin, and bufotenine) have also been found to ameliorate the symptoms associated with SLE (Figure 3). There has been growing interest to explore the anti-inflammatory or immunosuppressive activities of naturally derived phytoconstituents, as they have negligible toxicities and superlative efficiency [57]. The ensuing sections delineate the different categories of immunoregulatory and symptomatic relief providing phytoconstituents. 

## 4. Natural Sources of Immunoregulatory Drug Candidates

Phytocompounds have emerged as safer immunomodulatory therapeutics and have been used since time immemorial. These phytoconstituents serve as immunoregulatory drug candidates by means of either downregulating the functions of immune cells or by controlling the production of inflammatory cytokines, thereby maintaining immune homeostasis [35]. The main mechanism behind the action of a natural immunosuppressant is dependent on its antagonistic action on oxidative stressors, that ultimately leads to the formation of free radicals targeting the cytokines and autoantibodies. Such immunomodulation has been shown to be exerted by a number of plants, namely *Acacia farnesiana*, *Allium sativum*, *Andrographis paniculata*, *Angelica glauca*, *Arundo donax*, *Camellia sinensis*, *Commelina benghalensis*, *Coriandrum sativum*, *Costus speciosus*, *Curcuma longa*, *Cymbopogon citratus*, *Dracocephalum rupestre*, *Malus domestica*, *Morinda citrifolia*, *Ocimum gratissimum*, *Paeonia lactiflora*, *Picrorhiza scrophulariiflora*, *Salvia miltiorrhiza* and *Uncaria tomentosa*, etc. These plants contain one or more phytoconstituents that exhibit an exclusive antioxidant or anti-inflammatory activity. These phytoconstituents may also act as modulators of cytokine production and B–T cell co-stimulation, thereby eliminating the chances of the formation of autoantibodies and hyperactivation of the immune response in the case of SLE [58,59,60,61]. Further details of the category of phytoconstituents present in the immunosuppressive plants are elucidated in Table 1.

## 5. Phytoconstituents as Intracellular Signaling Regulators

Certain natural plant products are known to exhibit a strong anti-inflammatory activity by means of regulating the expression of various signaling protein kinases and cell cycle proteins which are involved in the transcription of inflammatory cytokines and autoantibodies. One such example is *Curcuma longa* (active phytoconstituent: curcumin) which is known to inactivate the transcription of TLR2 and NFκB, both of which are involved in preparing the plasmacytoid dendritic cell for presenting the autoantigens, thereby leading to the production of autoantibodies. Additionally, curcumin is also known to inhibit the expression of STAT3 and hence, downstream activation of proinflammatory cytokines, namely IL17 and BCL6, can be prevented [35,41]. In other studies, it has been found that *Tripterygium wilfordii* contains a diterpenoid epoxide, namely triptolide which is known to inhibit the transcription of IL-17 and STAT3 [35,39]. Similarly, an alkaloid, namely berbamine, found in *Berberis aristata*, is known to downregulate the expression of STAT3, and also inhibits the production of interferons that induce apoptotic cascade in SLE [15]. Several other plant species, such as *Andrographis paniculata, Argyrolobium roseum, Artemisia vestita, Bupleurum falcatum, Campylotropis hirtella, Clerodendron trichotomum, Dracocephalum kotschyi, Periploca sepium* and *Salvia mirzayanii*, have also been found to exhibit similar roles as intracellular signaling regulators and hence may possess probable utility as natural remedies for SLE [35,58,62,63,64].

## 6. Holistic Herbs as Symptomatic Relief Providers

Conventional medicines for treating SLE include the use of corticosteroids, antimalarial drugs and non-steroidal anti-inflammatory drugs (NSAIDS). However, these treatment regimens offer contingent results that may vary with changing patient profiles. Moreover, all these chemotherapeutic moieties are unsafe for continued use as they may induce organ toxicities [44,45,51]. Under such circumstances, alternative medicine treatments including the use of holistic herbs and natural plant products have gained a thriving interest as a viable remedial option for treating SLE. These holistic herbs have manifold targets and ultimately provide symptomatic relief in the case of SLE, which unfolds in several clinical manifestations and opportunistic ailments as well [35,57,65,66].

The primary clinical manifestations of SLE are cardiovascular, cerebrovascular, hematopoietic, hepatic, mucocutaneous, musculoskeletal, pulmonary and renal afflictions. Phytoconstituents, namely bufotenine (*Mucuna pruriens*), papaverine (*Papaver somniferum*), pseudoephedrine (*Ephedra sinica*), pyrocatechol (*Allium cepa, Achillea millefolium*), pyrogallol (*Anogeissus latifolia*), theophylline (*Camellia sinensis*), tyramine (*Crataegus laevigata*) and yohimbine (*Catharanthus roseus*), are known to exhibit symptomatic relief in cases of cardiovascular manifestations. Natural compounds such as tocopherol, arecoline, baicalin, ginkgolide, glabridin, naringenin, quercetin, rosmarinic acid, tangeretin and puerarin particularly, serve as cerebroprotective and neurostimulating remedies. Cynocobalamin and folacin (*Aloe vera, Anacardium occidentale, Angelica sinensis, Arachis hypogaea, Avena sativa, Beta vulgaris, Brassica rapa, Medicago sativa, Panax quinquefolius,* etc.) serve as hematopoietic and hepatoprotective remedies. Similarly, *Ephedra antisyphilitica, Alchemilla arvensis, Bixa orellana, Cimicifuga racemosa, Helianthus annuus, Ledum palustre, Piper longum,* etc., serve as remedies for managing renal and pulmonary afflictions. Most specifically, the maculopapular manifestation in the form of butterfly shaped cutaneous rashes can be treated with the aid of herbal drugs such as *Achyranthes aspera, Allium cepa, Allium sativum, Aloe vera, Azadirachta indica, Bauhinia variegata, Beta vulgaris, Calendula officinalis, Camellia sinensis, Curcuma longa, Lawsonia inermis, Lycopersicon esculentum, Momordica charantia, Rosmarinus officinalis* and *Thymus vulgaris* [22,35,58,62,63,67,68,69,70,71]. The precise understanding of the underlying mechanisms and the predominant phytoconstituents responsible for the remedial activity of these plants would need detailed preclinical and clinical research.

## 7. Propitious Drug Candidates

All the above mentioned plant-derived phyto-molecules exert their anti-SLE effect by one means or the other, such as (i) nitric oxide synthase (iNOS) and COX-2 inhibitors; (ii) autoantibody production inhibitors; (iii) lymphocyte proliferation inhibitors; (iv) NFκB and NFAT-mediated signaling inhibitors; (v) STAT3, CREMα and Rho-associated protein kinase (ROCK) inhibitors; (vi) pro-inflammatory cytokine production regulators; and (vii) symptomatic relief providers. Some of the natural plant products also provide manifold therapeutic benefits as they act upon more than one of the molecular targets of SLE. One such example is *Allium sativum* which is known to maintain immune system homeostasis by inhibiting Th1 proinflammatory cytokines. Moreover, it also exerts an inhibitory effect on NFκB activation. It has also been proposed as a topical remedy for SLE manifestations, including butterfly shaped cutaneous lesions and oral ulcers [34].

Furthermore, *Andrographis paniculata*, *Camellia sinensis*, *Coriandrum sativum* and *Curcuma longa* serve as immunomodulatory as well as anti-inflammatory herbs, wherein they probably act via reducing the levels of iNOS, COX-2, PGE2, TNF-alpha, IL-6 and IL-12. Hence, these herbs may aid in suppressing inflammation by inhibiting the proliferation of pro-inflammation cytokines, as well as impeding the activation of NFκB signaling [24,35,36,38,41].

It is noteworthy that certain phyto-molecules such as alizarin, asperuloside, chrysophanol, digoxin and berbamine isolated from *Berberis aristata* and *Morinda citrifolia* serve as specific signaling regulators for downregulating the expression of STAT3, RAF1 and mTOR. Hence, both these herbals may serve as promising sources of lead drug molecules targeting SLE [15,42].

These suggested herbal drug candidates are devoid of any potential toxicity or adverse drug reactions, as evident from their previously published toxicological profiles [57]. Henceforth, looking upon the proposed medicinal utility and negligible toxicities, it is postulated that these herbal metabolites should be tested for clinical development and further human use.

## 8. Conclusions

Systemic lupus erythromatosus is a prevalent autoimmune disorder with life threatening consequences. It causes widespread cutaneous as well as systemic inflammation, thereby leading to multiple organ damage. There is still a dearth of absolute treatment regimens for the appropriate management of this disease, wherein the available synthetic chemotherapeutic drugs possess multiple toxicities. As a consequence, one must utilize the alternative therapeutic modalities in the form of natural plant products as these phytoconstituents have negligible toxicities, if any, and are endowed with holistic therapeutic activities. These phytoconstituents may serve as immunoregulatory moieties, intracellular signaling regulators and symptomatic relief providers. Certain herbals like *Allium sativum*, *Angelica glauca*, *Camellia sinensis* and *Curcuma longa* may serve to cater manifold benefits as they can aid in both preventing the occurrence of SLE by acting as immunomodulators and managing the consequences of SLE by providing symptomatic relief. These suggested phyto-metabolites act via regulating a specific molecular target and hence can serve as probable drug candidates for developing a suitable pharmacophore. Undoubtedly, more research must be performed to evaluate the mechanistic roles and clinical benefits of these natural plant products before engaging them as the alternative treatment armamentarium of SLE.

## Figures and Tables

**Figure 1 cells-09-01049-f001:**
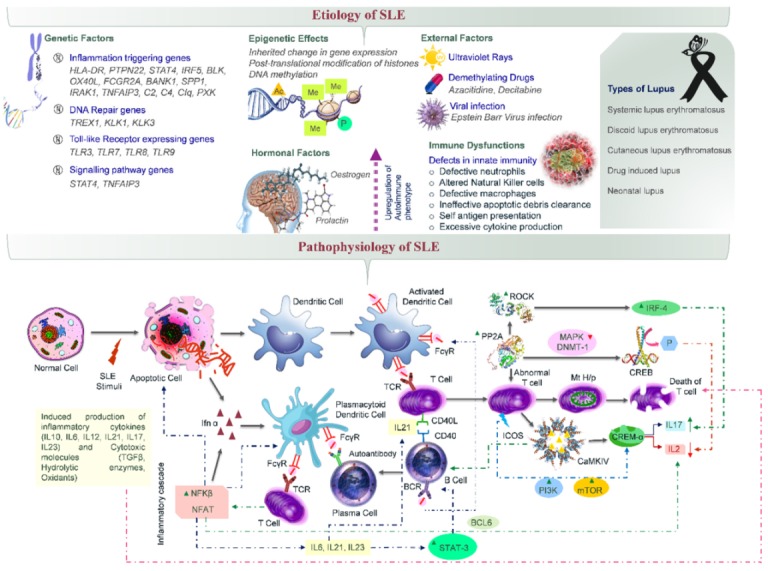
Etiology and pathophysiology of systemic lupus erythromatosus (SLE). Etiology of SLE includes genetic, epigenetic, hormonal, immunological and other external factors. These etiological factors act as a stimulus for inducing apoptosis of a normal cell, thereby breaking the dsDNA which in turn induces the production of interferons (Ifn-α) [1,10]. These interferons induce the activation of plasmacytoid dendritic cells. Further, the broken DNA acts as an antigen which is presented on dendritic cells (physiologically altered with FcγR). Meanwhile, T cells and B cells also interact via CD40 and CD40L interaction with the DNA antigen, thereby inducing the production of autoantibodies which further induce the interaction of plasmacytoid dendritic cells and T cells [12,13,14]. This interaction upregulates the production of NFAT and NFκB, both of which inhibit the production of IL-2 (an immunosuppressive cytokine) and induce the production of BCL6, IL6, IL21 and IL23. These cytokines further upregulate the expression of STAT3 which aids in the co-stimulation of B cells and T cells, thereby leading to the production of autoantibodies [13,14,15]. These autoantibodies induce SLE-related manifestations and a simultaneous signaling cascade also commences. Protein phosphatase 2 (PP2A) initiates a 3-tier phenomenon [16], i.e.: mitochondrial hyperpolarization and the death of T cells [17,18,19]; activation of Rho-associated protein kinase (ROCK) which further mediates the binding of IL17 transcription enhancer interferon regulatory factor 4 (IRF4) [20]; dephosphorylation of cAMP-responsive element-binding protein 1 (CREB), resulting in suppression of IL2 transcription (mediated by downregulation of expression of MAPK—mitogen-activated protein kinase and DNMT1—DNA methyl transferase 1) [21]. Further upregulation of IL17 and downregulation of IL2 production is mediated by calcium/calmodulin-dependent protein kinase IV (CaMKIV), which in turn increases the binding of cAMP response element modulator (CREMα) [12,13,14,15,16,17,18]. Certain co-stimulatory signaling molecules such as ICOS (Inducible T cell co-stimulator), PI3K (phosphoinositide 3-kinase) and mTOR (mechanistic target of Rapamycin) also aid in similar interleukin regulation [12].

**Figure 2 cells-09-01049-f002:**
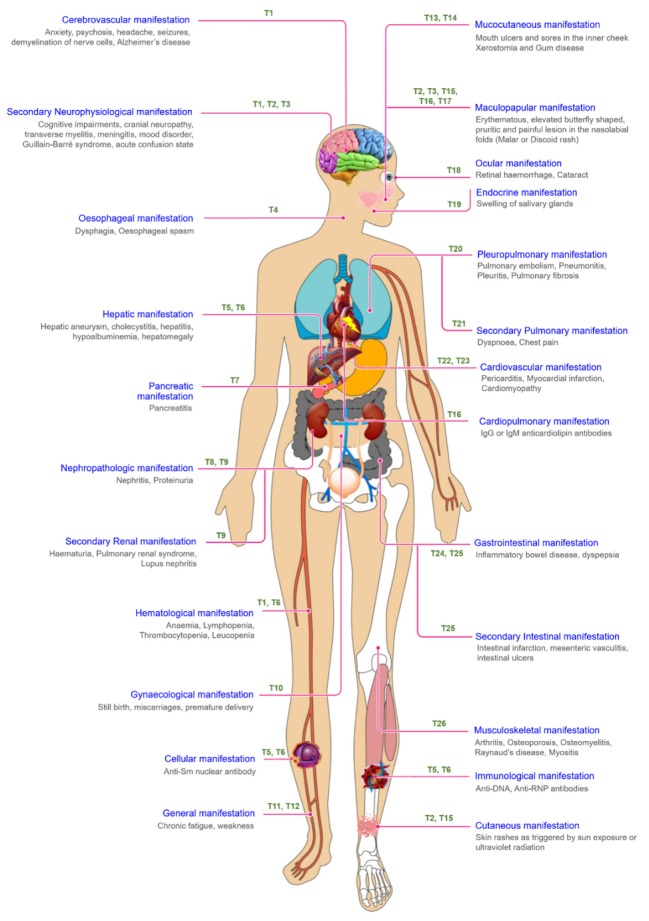
Clinical manifestations of systemic lupus erythromatosus along with therapeutic interventions. Systemic lupus erythromatosus is a chronic autoimmune disease targeting various systems of the human body including the cardiovascular, cerebrovascular, gastrointestinal, gynecological, hematological, hepatic, immunological, mucocutaneous, musculoskeletal, ocular, esophageal, pancreatic, pulmonary and renal systems. The treatment of SLE is centered upon formulating a regimen of topical and systemic therapies designed to reduce both disease severity and activity, wherein various treatment modalities are T1: dapsone and fingolimod; T2: lenalidomide; T3: thalidomide; T4: calcium channel blockers; T5: glucocorticoid prednisone; T6: azathioprine; T7: rifampin; T8: chaperonin 10; T9: mycophenolate mofetil; T10: misoprostol; T11: iron supplements; T12: vitamin B complex; T13: benzocaine; T14: salbutamol; T15: clofazimine; T16: IVIG (intravenous immunoglobulin); T17: cefuroxime; T18: quinacrine; T19: NSAIDs (non-steroidal anti-inflammatory drugs); T20: methylprednisolone; T21: ipatropium bromide; T22: ibuprofen; T23: aspirin; T24: corticosteroids; T25: aminosalicylate; and T26: methotrexate.

**Figure 3 cells-09-01049-f003:**
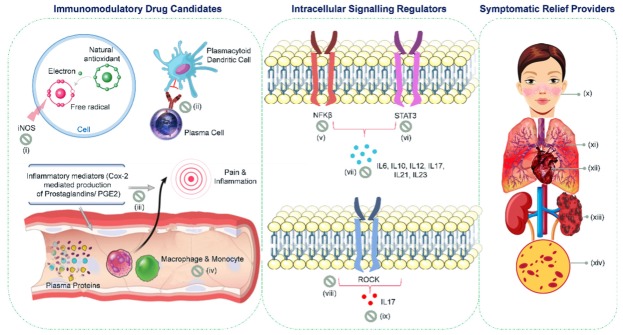
Pharmaco-mechanistic profile of phyto-metabolites in treating systemic lupus erythromatosus. Plant-derived phytoconstituents serve as (**i**) nitric oxide synthase (iNOS) inhibitors and promote the scavenging of free radicals, thereby preventing SLE associated DNA damage (alizarin, asperuloside, α-amyrin, β-amyrin, andrographolide, angelic acid, cadinol, isophytol, β-phellandrene, galic acid, kaempferol, limonene, lupeol, lectin, scrocaffeside A, salvianolic acid, mitraphylline); (**ii**) inhibitors of autoantibody production {linalool acetate, α-terpinyl acetate}; (**iii**) inhibitors of COX-2-mediated PGE2 production, thereby ameliorating inflammation and pain {andrographolide, isophytol, farnesene, cadinoleugenol, troptolide}; (**iv**) inhibitors of lymphocyte proliferation {limonene, carvacrol, terpinene, calycopterin}; (**v**) down-regulators of NFκB -mediated signaling {allicin, alliin-γ-glutamyl-S-allyl-L-cysteine, andrographolide, saikosaponin, epigallocatechin gallate, lupeol, taraxerol, friedelin, betulinic acid, linalool, pinene, terpinene, limonene, curcumin}; (**vi**) down-regulators of STAT3-mediated signalling {berbamine, curcumin}; (**vii**) inhibitors of production of proinflammatory cytokines {allicin, alliin-γ-glutamyl-S-allyl-L-cysteine, pinitol, theophylline, epigallocatechin gallate, curcumin, periplocoside E, triptolide}; (**viii**) down-regulators of the expression of Rho-associated protein kinase (ROCK) {curcumin}; (**ix**) inhibitors specifically of the production of IL-17 {curcumin, quinone, phenylpropanoids, steroids, periplocoside E}; (**x**) a topical remedy for butterfly shaped cutaneous lesions {ecdysterone, oleanolic acid, pyrocatechol, allicin, aloin, aloe emodin, β carotene, β sitosterol, azadirachtin, cynocobalamin, Plastoquinone, xanthone, coumarin, lycopenecurcubitacin, rosmarinic acid}; (**xi**) a pulmonary protectant remedy {bixin, ledol, ascaridole}; (**xii**) a cardioprotectant remedy {tyramine, pseudoephedrine, bufotenine, papaverine}; (**xiii**) a nephroprotectant remedy {cimicifugic acid, protocatechuic acidephedrineferulic acid, gallic acidchavicine, piperine}; (**xiv**) a hematopoietic remedy {cynocobalamin, folacin, flavolignans, β carotene, ginsenoside, etc.}.

**Table 1 cells-09-01049-t001:** Major phyto-metabolites along with their targeted utility as therapeutic potentials for systemic lupus erythromatosus.

Molecular Target	Probable Utility	Therapeutic Benefits	Source Plant	Active Metabolites #	Reference (s)
iNOS and COX	Decreasing the nitrite and PGE2 levels by inhibition of the iNOS and COX expression	Anti-inflammatory	*Acacia farnesiana*	Lectin, Lupeol, α-Amyrin, β-Amyrin	[23]
			*Andrographis paniculata*	Andrographolide	[24]
			*Angelica glauca*	Angelic acid, β-Phellandrene, Ligustilide, Limonene	[25]
			*Arundo donax*	N-acetyl-D-glucosamine lectin	[26]
			*Malus domestica*	Isophytol, Farnesene, Cadinol	[27]
			*Ocimum gratissimum*	Eugenol	[28]
			*Paeonia lactiflora*	Gallic acid, Kaempferol	[29]
			*Picrorhiza scrophulariiflora*	Caffeoyl glycosides, Phenylethanoid glycoside, Plantamajoside, Scrocaffeside A	[30]
			*Salvia miltiorrhiza*	Salvianolic acid, Dihydrotanshinone, Tanshinone	[31]
			*Uncaria tomentosa*	Mitraphylline	[32,33]
NFκB	Inhibits T cell activation through the modulation of NFκB transcription factor; reducing the level of pro-inflammatory cytokines	Immunomodulator, Signaling Regulator	*Allium sativum*	Allicin, Alliin, γ-Glutamyl-S-allyl-L-cysteines	[34]
			*Bupleurum falcatum*	3-O-Feruloyl 5-O-Caffeoylquinic acid, Saikosaponin	[35]
			*Clerodendron trichotomum*	Lupeol, Friedelin, Betulinic acid, Taraxerol	[35]
			*Coriandrum sativum*	Linalool, Terpinene, Pinene, Limonene, p-Cymene	[36]
Th1/Th2 proinflammatory cytokines	Suppress inflammation, inhibit proliferation and pro-inflammatory cytokines, downregulation of Th1/Th2 cytokines expression	Immunomodulator	*Argyrolobium roseum*	Pinitol	[37]
			*Camellia sinensis*	Theophylline, Epigallocatechin gallate	[38]
			*Tripterygium wilfordii*	Triptolide	[35,39,40]
STAT3 and ROCK	Downregulating the expression of STAT3 and ROCK	Signaling regulator	*Berberis aristata*	Berbamine	[15]
			*Curcuma longa*	Curcumin	[35,41]
*RAF1* and *mTOR*	Downregulating the expression of V-raf-leukemia viral oncogene 1 (*RAF1*), and mechanistic target of rapamycin (*mTOR*) mRNA	Signaling regulator	*Morinda citrifolia*	Alizarin, Asperuloside, Chrysophanol, Digoxin	[42]

# Active phytoconstituents that may serve as phyto-ligands for assessing the structure–activity relationships, thereby generating a pharmacophore.

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
