# Peer review of "Mechanistic Paradigms of Natural Plant Metabolites as Remedial Candidates for Systemic Lupus Erythromatosus"

_cells, 2020, doi:10.3390/cells9041049_

Round 1

Reviewer 1 Report

In this review, Acharya Balkrishna et al., reports the interest of some phytomolecules in the treatments (preventive and curative) of SLE patients. Although this review is interesting, some additions have to be performed to render it more attractive and robust.

  1. Figure 1. The list of signaling pathways suggested being involved in lupus etiology has to be supported by publications (references).
  2. In the “classical” regimens of SLE patients, authors should include the most recent ones including anti-CD20 (rituximab) and anti-BAFF mAbs
  3. Page 4, the authors mixed transcription factors with kinases, please could they organize the cell signaling part and avoid to mix these factors.
  4. Instead of a list of phytomolecules targeting different enzymes involved in SLE pathogenesis, authors should combine these molecules based on a common molecular target because this could help computer modelers to isolate pharmacophore if the molecules target the same region within the enzyme.

Author Response

Query 1: Figure 1. The list of signalling pathways suggested being involved in lupus etiology has to be supported by publications (references).

Reply: Thank you very much for these suggestions, we have now incorporated appropriate references for all the listed pathways involved in lupus etiology. The same references have been highlighted in the figure legends.

Query 2: In the ‘classical’ regimens of SLE patients, authors should include the most recent ones including anti-CD20 (rituximab) and anti-BAFF mAbs.

Reply: Our apologies for the hindsight. As suggested, we have now incorporated both the recommended treatment regimes, namely, anti-CD20 (rituximab) and anti-BAFF mAbs in the manuscript.

Query 3: Page 4, the authors mixed transcription factors with kinases, please could they organize the cell signalling part and avoid to mix these factors.

Reply: This error is regretted. The categorization of the transcription factors and nuclear factors have now been clearly mentioned, thereby out-ruling the ambiguity of transcription factors and kinases.

Query 4: Instead of a list of phytomolecules targeting different enzymes involved in SLE pathogenesis, authors should combine these molecules based on a common molecular target because this could help computer modellers to isolate pharmacophore if the molecules target the same region within this enzyme.

Reply: This is indeed an excellent suggestion. Molecular target for all the suggested phytomolecules have now been added in Table 1, as suggested, thereby providing probable candidates for pharmacophore generation. The possibility of generation of a pharmacophore has also been highlighted in the conclusion section, with suggestions for further research.

Reviewer 2 Report

Because limited knowledges regarding herbal medicine in the treatment of patients with SLE are available, the review article is somewhat interesting enough for readers of the journal. However, the description in the current form remains too rough to easily understand core tip of the MS.

1.     The authors describe so many herbal agents as possible therapeutic candidates for treatment of SLE. Thus, it is hard to easily understand possible efficacy of each herbal medicine for treatment of SLE. Some representative candidates should be chosen and discussed in more-depth their potential mechanisms against the pathogenesis of SLE.

2.     Also, efficacy and safety of representative herbal agents in the management of SLE in clinical practice should be clearly described.

3.     It is nice to focus on most hopeful herbal agents at this time should be represented. This issue may depend on the authors’ opinion.

Author Response

Query 1: The authors describe so many herbal agents as possible therapeutic candidates for treatment of SLE. Thus, it is hard to easily understand possible efficacy of each herbal medicine for treatment of SLE. Some representative candidates should be chosen and discussed in more-depth their potential mechanisms against the pathogenesis of SLE.

Reply: Thank you very much indeed for this advice. The provided list of herbals has now been theoretically categorized with respect to their specific molecular targets (Table 1 remodified). Moreover, representative herbal candidates possessing manifold benefits have now been discussed in detail, along with their potential mechanism against the pathogenesis of SLE (Described under the section head 7. Propitious Drug Candidates).

Query 2: Also, efficacy and safety of representative herbal agents in the management of SLE in clinical practice should be clearly described.

Reply: Thank you again. The efficacy and safety profile of the selected representative herbal agents have now been clearly mentioned in the text (Last paragraph of section 7. Propitious Drug Candidates), thereby clarifying their potential safe use in clinical practice. 

Query 3: It is nice to focus on most hopeful herbal agents at this time should be represented. This issue may depend on the authors’ opinion.

Reply: Thank you. As suggested, the most hopeful herbs with manifold therapeutic benefits have now been highlighted both in the table (modified Table 1), and text (altogether a different section formulated).

We appreciate your in-depth thoughts and advice on this review article. With these modifications, the manuscript indeed reads better and clearer. Thank you very much once again.

Round 2

Reviewer 1 Report

The authors addressed all of the raised concern, thereby I think that this review could be very attractive and interesting for the readers of Cells.

Reviewer 2 Report

The revised MS is much improved.